# Restricted Water Diffusion in Diffusion-Weighted Magnetic Resonance Imaging in Pancreatic Cancer is Associated with Tumor Hypoxia

**DOI:** 10.3390/cancers13010089

**Published:** 2020-12-30

**Authors:** Philipp Mayer, Anne Kraft, Hagen R. Witzel, Nicole Marnet, Nina Hörner, Wilfried Roth, Stefan Heinrich, Thilo Hackert, Frank Bergmann, Hans-Ulrich Kauczor, Miriam Klauss, Matthias M. Gaida

**Affiliations:** 1Clinic for Diagnostic and Interventional Radiology, University Hospital Heidelberg, 69120 Heidelberg, Germany; Hans-Ulrich.Kauczor@med.uni-heidelberg.de (H.-U.K.); Miriam.Klauss@med.uni-heidelberg.de (M.K.); 2Institute of Pathology, University Medical Center Mainz, JGU-Mainz, 55131 Mainz, Germany; Anne.Kraft@unimedizin-mainz.de (A.K.); Hagen.Witzel@unimedizin-mainz.de (H.R.W.); Nicole.Marnet@unimedizin-mainz.de (N.M.); Nina.Hoerner@unimedizin-mainz.de (N.H.); Wilfried.Roth@unimedizin-mainz.de (W.R.); Matthias.Gaida@unimedizin-mainz.de (M.M.G.); 3Department of Surgery, University Medical Center Mainz, JGU-Mainz, 55131 Mainz, Germany; Stefan.Heinrich@unimedizin-mainz.de; 4Department of General, Visceral, and Transplantation Surgery, University Hospital Heidelberg, 69120 Heidelberg, Germany; Thilo.Hackert@med.uni-heidelberg.de; 5Institute of Pathology, University Hospital Heidelberg, 69120 Heidelberg, Germany; Frank.Bergmann@mail.klinikum-darmstadt.de; 6Clinical Pathology, Klinikum Darmstadt GmbH, 64283 Darmstadt, Germany; 7Research Center for Immunotherapy, University Medical Center Mainz, JGU-Mainz, 55131 Mainz, Germany; 8Joint Unit Immunopathology, Institute of Pathology, University Medical Center, JGU-Mainz and TRON, Translational Oncology at the University Medical Center, JGU-Mainz, 55131 Mainz, Germany

**Keywords:** pancreatic ductal adenocarcinoma, hypoxia, diffusion-weighted imaging

## Abstract

**Simple Summary:**

Pancreatic cancer is characterized by a dense network of connective tissue surrounding clusters of cancer cells, the so-called stroma. This ubiquitous connective tissue impairs the delivery of oxygen to cancer cells. This results in hypoxia, which renders the cancer more aggressive and more resistant to treatment. In the present study, we investigated whether the extent of hypoxia in pancreatic cancer can be predicted by magnetic resonance imaging (MRI), a widely used medical imaging technique. More specifically, we used an MRI sequence which can quantitate the random motion (i.e., diffusion) of water molecules within the cancer tissue, namely diffusion-weighted (DW) MRI. We found that the random motion of water molecules is lower in cancer lesions with high hypoxia compared to those with low hypoxia. The findings from our study imply that DW-MRI can be used to identify pancreatic cancer lesions with high hypoxia which are at high risk for treatment failure.

**Abstract:**

Hypoxia is a hallmark of pancreatic cancer (PDAC) due to its compact and extensive fibrotic tumor stroma. Hypoxia contributes to high lethality of this disease, by inducing a more malignant phenotype and resistance to radiation and chemotherapy. Thus, non-invasive methods to quantify hypoxia could be helpful for treatment decisions, for monitoring, especially in non-resectable tumors, or to optimize personalized therapy. In the present study, we investigated whether tumor hypoxia in PDAC is reflected by diffusion-weighted magnetic resonance imaging (DW-MRI), a functional imaging technique, frequently used in clinical practice for identification and characterization of pancreatic lesions. DW-MRI assesses the tissue microarchitecture by measuring the diffusion of water molecules, which is more restricted in highly compact tissues. As reliable surrogate markers for hypoxia, we determined Blimp-1 (B-lymphocyte induced maturation protein), a transcription factor, as well as vascular endothelial growth factor (VEGF), which are up-regulated in response to hypoxia. In 42 PDAC patients, we observed a close association between restricted water diffusion in DW-MRI and tumor hypoxia in matched samples, as expressed by high levels of Blimp-1 and VEGF in tissue samples of the respective patients. In summary, our data show that DW-MRI is well suited for the evaluation of tumor hypoxia in PDAC and could potentially be used for the identification of lesions with a high hypoxic fraction, which are at high risk for failure of radiochemotherapy.

## 1. Introduction

Pancreatic ductal adenocarcinoma (PDAC) is a highly aggressive cancer with an unfavorable five-year survival rate of about nine percent [1]. Hypoxia is emerging as a key factor of the aggressive tumor biology and the relative resistance to conventional as well as targeted therapies [2]. Compared to other solid tumors, oxygen levels in PDAC are among the lowest [3,4]. Several factors contribute to the low oxygen tension. Among these are its characteristically low microvascular density [5], rapid tumor growth which may result in impaired architecture of the tumor vessels, as well as the extensive desmoplastic stroma that exerts mechanical stress on the tumor vasculature and thus impairing tissue perfusion [6]. In PDAC cells, hypoxia induces the expression of a number of genes, particularly tumor-promoting transcription factors such as the hypoxia-inducible factors (HIFs), the master regulator of cell adaption to hypoxia [3]. Reduced oxygen levels and consequent stabilization of HIF-1 facilitate stromal remodeling by activation of pancreatic stellate cells [7] to produce matrix molecules, including type-I collagen [8], periostin, and fibronectin [9]. Thus, hypoxia and the dense tumor stroma of PDAC are interdependent and may potentiate each other in a positive feedback loop [2], termed hypoxia-fibrosis cycle [9]. Via a switch from oxidative phosphorylation to glycolysis, HIF-1 enables cancer cells to survive in hypoxic conditions [6]. HIF-1 induces the conversion of conventional pancreatic cancer cells into cells with stem cell-like features [10]. The latter represent a distinct cell population that is involved in resistance to standard treatments [11]. Hypoxia, which is not only found in the tumor center, but also at the tumor invasion front [12], promotes cancer cell invasion and metastasis by increasing the formation of invadopodia [13] and by inducing epithelial–mesenchymal transition (EMT) in a HIF-1-dependent manner [2]. Moreover, hypoxia induces via HIF-1 the upregulation of VEGF in pancreatic cancer, where high VEGF expression is associated with aggressive disease and increased metastasis formation [14,15]. Another recently established surrogate marker for intratumor hypoxia in PDAC is Blimp-1. At first discovered as a main regulator controlling the survival and differentiation of B cells [16], Blimp-1 has recently evolved to be aberrantly expressed in different tumors such as PDAC, induced by various cytokines and also hypoxia and promoting tumor cell invasion and metastasis [17,18]. Recent studies reported high levels of Blimp-1 to be associated with a poor prognosis in abdominal cancers, among these PDAC [18,19], which emphasizes the biological relevance of Blimp-1 in cancer progression and enforces deeper studies.

Hypoxia not only induces a malignant and invasive phenotype of PDAC cells, but also renders the cells more resistant to radiation and chemotherapy [3,20]. The resistance is partly conferred by HIF-1 and its downstream effectors, such as VEGF; for the latter, therapeutic interventions are currently under clinical investigation [3,21]. Under hypoxic conditions, the cytotoxicity of various chemotherapeutic agents whose activity is dependent on generating free radicals including active oxygen species is reduced [22]. Several therapeutic strategies targeting the hypoxic microenvironment have been investigated, among these hypoxia-activated prodrugs [23]. Hence, assessment of the hypoxic fraction in PDAC has a potential impact on treatment stratification and could improve therapy outcome in the near future [6]. This urges for non-invasive imaging techniques as a surrogate marker for hypoxia in PDAC.

Several non-invasive imaging techniques have been explored over the past years for non-invasive quantification of tumors’ oxygenation status, each with different advantages and drawbacks. Positron emission tomography (PET) with the tracer 18F-fluoromisonidazole (FMISO) is one of the most extensively investigated methods with theoretically high specificity for detection of hypoxic tissue, but is limited by its high costs and limited availability [24]. Increasing evidence suggests that magnetic resonance imaging (MRI) techniques may provide a practical and more readily available alternative for hypoxia imaging [25]. Blood oxygenation-level dependent (BOLD) MRI makes use of paramagnetic properties of deoxyhemoglobin and is mainly used for regional quantification of oxygenation of the brain [25]. Long scanning times due to its intrinsic low sensitivity and proneness to motion artifacts, however, complicate its use in the abdomen [26]. Another functional MRI technique, termed diffusion-weighted (DW)-MRI, is not limited by these drawbacks of BOLD imaging and is widely used in clinical practice for probing the microarchitecture of abdominal organs [27]. DW-MRI quantitates the random Brownian motion of water molecule protons in human tissues by a parameter termed apparent diffusion coefficient (ADC). Diffusion in PDAC was shown to be dependent on several tissue properties including tumor-stroma ratio [28] and immune cell infiltration [29,30]. In general terms, diffusion is usually more restricted in compact tissues [27] which are also prone to developing hypoxia [22]. Previous studies indicated the suitability of DW-MRI as a surrogate parameter for oxygenation in melanoma and prostate cancer tissue [31,32]. However, studies on the use of DW-MRI to quantitate hypoxia in PDAC are lacking.

In the present study, we aimed to investigate whether ADC from DW-MRI can be used as a patient-based non-invasive surrogate for tumor hypoxia in PDAC.

## 2. Results

### 2.1. Diffusion-Weighted MRI and Correlation to Histologic Data

Diffusion-weighted MRI scans were conducted in 42 patients with final histopathological diagnosis of a PDAC on the day before surgery. Two board-certified radiologists analyzed the diffusion-weighted images (DWI). A monoexponential diffusion model was applied and ADC values were quantitated. ADC values varied among patients. Agreement was good for ADC values (intraclass correlation coefficient (ICC) = 0.840). In the following, ADC values are provided as mean values for both radiologists unless specified differently.

ADC values ranged from 0.827 to 1.509 µm^2^/s (median 1.215 µm^2^/s, interquartile range (IQR) 1.054 to 1.337 µm^2^/s). These ADC values are similar to ADC values from prior studies on DW-MRI in PDAC [33,34]. In the corresponding tissue samples, the expression of Blimp-1 and VEGF was analyzed by immunohistochemistry and quantified by the Allred score. Blimp-1 Allred scores ranged from 0 to 7 (median 3, IQR 2 to 4); VEGF Allred scores ranged from 0 to 8 (median 4, IQR 3 to 7). Blimp-1 showed nuclear expression and was detected in 40 out of 48 samples. VEGF showed cytoplasmatic expression and was detected in 43 out of 47 samples. We observed negative significant Spearman rank-correlations between ADC values and Blimp-1 Allred score (r_s_ = −0.674, *p* < 0.001) as well as between ADC values and VEGF Allred score (r_s_ = −0.561, *p* < 0.001). There was a significant positive rank-correlation between Blimp-1 Allred score and VEGF Allred score (r_s_ = 0.391, *p* = 0.011). Median Blimp-1 and VEGF Allred scores were significantly higher in patients with low ADC values (≤1.2 µm^2^/s) compared to patients with high ADC values (>1.2 µm^2^/s) (median score 4 versus 2 for Blimp-1, *p* < 0.001; median score 7 versus 4 for VEGF, *p* < 0.001; Mann–Whitney test). Vice versa, according to the Mann–Whitney test, patients with low Blimp-1 Allred scores (<3) or low VEGF Allred scores (<5) had significantly higher median ADC values than those with high Blimp-1 expression (Allred scores ≥3) or high VEGF expression (Allred scores ≥5) (median ADC 1.350 µm^2^/s versus 1.143 µm^2^/s for Blimp-1, *p* < 0.001; median ADC 1.292 µm^2^/s versus 1.0741 µm^2^/s, *p* < 0.001) (Figure 1 and Appendix A). The median tumor-stroma ratio was 30:70 in the present study. ADC values as well as Blimp-1 and VEGF Allred scores did not differ significantly between stroma-rich tumors (stroma content >70%) and stroma-poor versus (stroma content ≤70%) (median ADC 1.306 µm^2^/s versus 1.192 µm^2^/s, *p* = 0.212; median Blimp-1 Allred score 3 versus 3, *p* = 0.590; median VEGF Allred score 4 versus 4.5, *p* = 0.322; Mann–Whitney test). Blimp-1 and VEGF Allred scores, tumor-stroma ratios, as well as ADC values are listed in Appendix A.

### 2.2. Regulation of Blimp-1 under Hypoxic Conditions in Pancreatic Tumor Cells

To assess a molecular link between hypoxia and Blimp-1 expression and to provide evidence for its reliability as a hypoxic marker, pancreatic tumor cells were cultivated in a medium containing Cobalt(II)chloride (CoCl_2_), a widely used and well-established molecular model mimicking hypoxia in cell culture [35]. Two different pancreatic cancer cell lines were used because due to their different origin and differentiation status, their response to hypoxia might vary. In both cell lines, an upregulation of Blimp-1 (*n* = 6) was detected at the protein level, but more pronounced in AsPc1 (2.01-fold, *p* = 0.025, paired samples t-test) and very slight—though reproducible—in T3M4 (1.12-fold, *p* = 0.527, paired samples t-test) compared to the untreated group (Figure 2). Further experiments indicated a possible involvement of the ERK pathway in Blimp-1 regulation (Appendix A), as previously described [18].

### 2.3. Regulation of Blimp-1 under Hypoxic Conditions in a Human Pancreatic Ex Vivo Tissue Culture

To corroborate the in vitro data, human pancreatic cancer tissue was used in an ex vivo culture system. Tissue slices of human pancreatic cancer samples, generated from surgical specimens of individual patients (*n* = 5) were treated for 24 h with 200 µM CoCl_2_ to induce a hypoxic micromilieu. Upregulation of Blimp-1, as a reliable marker of hypoxia, in the nucleus could be detected in 4 out of the 5 samples (*p* = 0.037, paired samples t-test) by immunohistochemistry (Figure 3). The median Allred immunoreactivity score in the untreated group was 4/8, in the treated group 6/8.

### 2.4. Regulation of Blimp-1 under Hypoxic Conditions in an Avian Xenograft Model

The avian xenograft model provides the possibility to induce hypoxia in an established tumor outgrow of human cells. The tumors were incubated twice with a filter paper containing CoCl_2_. After tumor resection, Blimp-1 protein expression was determined by immunohistochemistry. A significant upregulation could be detected in AsPc1 (*p* = 0.047, unpaired samples t-test), whereas no significant changes were observed for the cell line T3M4 (*p* = 0.384, unpaired samples *t*-test) (Figure 4). No differences in tumor size and weight could be established.

### 2.5. Analysis of Progression-Free Survival

Progression-free survival (PFS) was significantly shorter in patients with high Blimp-1 Allred scores (≥3) than in patients with low Blimp-1 Allred scores (<3, median 305 days versus 1434 days, *p* = 0.020, log-rank test). There was a trend towards shorter PFS in patients with low ADC values (≤1.2 µm^2^/s) compared to patients with high ADC values (>1.2 µm^2^/s, median 305 days versus 433 days, *p* = 0.282, log-rank test) which is in line with a recent study [36]. Similarly as reported by Ellis et al. [37], PFS of patients with high VEGF Allred scores (≥5) was not significantly different from PFS of patients with high VEGF Allred scores (<5, median 394 days versus 433 days, *p* = 0.625, log-rank test). Kaplan–Meier curves are displayed in Appendix A. Data on PFS are listed in Appendix A.

## 3. Discussion

During recent years, diffusion-weighted imaging (DWI) has developed into a useful adjunct for the diagnosis, characterization, and therapy monitoring of various abdominal malignancies using magnetic resonance imaging [27]. In addition to the morphological information from conventional MR sequences, DWI adds functional information by enabling the visualization of random Brownian motion of water molecules within a voxel of tissue [38,39]. Since the random motion, i.e., diffusion, of water molecules is dependent on tissue properties, DW-MRI offers a non-invasive probe for tissue microstructure [39]. In clinical practice, water diffusion is routinely measured by the so-called apparent diffusion coefficient (ADC). Most malignant tumors, including PDAC, typically exhibit restricted water diffusion, as expressed by low ADC values, when compared to surrounding non-neoplastic tissues [27]. In the majority of tumor entities, restriction of water diffusion is thought to be primarily the consequence of high cellularity with numerous cell membranes and reduced extracellular spaces [40]. In PDAC, there is also an inverse correlation between cellularity and ADC values [41]. However, since the non-neoplastic pancreas which is composed of densely packed cells appears to have a higher cell relative density than most PDAC lesions, which may have low cellularity, cell density cannot be the only contributor to the lower ADC values of PDAC lesions compared to the non-neoplastic pancreas [38]. One other potential determinator of diffusivity in PDAC is the tumor-stroma ratio with the collagenous fibers and other matrix molecules representing physical barriers for the diffusion of water molecules [28,42]. Both stroma content and cell density presumably contribute to the notoriously low oxygen tension in PDAC. The desmoplastic tumor stroma can impair oxygen supply by exerting mechanical stress on tumor vessels and thus diminishing tissue perfusion [6], while a higher cell density promotes hypoxia by increasing oxygen consumption [43]. Hypoxia, in turn, can contribute to PDAC progression by promoting invasiveness and metastasis formation [17,44].

The present study aimed to investigate whether the ADC can be used as a patient-based non-invasive surrogate for tumor hypoxia in PDAC. First, we searched for a hypoxic marker in human PDAC. A previous study by Chiou et al. identified the transcription factor Blimp-1 as a promising hypoxia marker in PDAC in vitro [17]. Blimp-1 was at first discovered as the main regulator controlling the survival and differentiation of B lymphocytes [16], but is now seen as a powerful regulator not only of B lymphocytes, but also of other cells of the immune system [45,46]. Chiou et al. showed that, in murine and human PDAC cell lines, Blimp-1 is induced rapidly after exposure to hypoxia and that, in a mouse model, Blimp-1 is apparently a major driver for metastasis formation [17], which is in line with our previous data showing that Blimp-1 enhances the invasive capacity of tumor cells [18].

One aim of the present study was to confirm the suitability of Blimp-1 as hypoxic marker in PDAC. To mimic hypoxia, we made use of a well-established and widely used hypoxia model, cobalt chloride (CoCl_2_)-induced chemical hypoxia [35]. Cobalt chloride stabilizes HIF-1α/2α, thus increasing the levels of HIF-1α/2α in a dose- and time-dependent manner, similarly as observed in conventional hypoxia, and increasing the transcription levels of downstream effector proteins [35] with one important downstream target of HIF-1α in PDAC being Blimp-1 [17]. Upon treatment with cobalt chloride, we observed an upregulation of Blimp-1 in PDAC cell lines, in tumor xenograft models, and much important in real human cultivated tumor tissue from surgical specimens in a 3D model. The degree of Blimp-1 upregulation varied between PDAC cell lines which could be explained by different susceptibilities of these cell lines to hypoxia [47,48,49].

Incubation of PDAC xenograft tumors with cobalt chloride revealed no changes of tumor volumes and weights when compared to vehicle controls. This indicates that hypoxia, and consequent Blimp-1 activation, does not increase cancer cell proliferation to a relevant extent. Together with previous xenograft experiments by us where PDAC cells treated with IL-21, a potent inductor of Blimp-1, exhibited enhanced migration [18], these results point towards Blimp-1 being primarily a cell migration factor boosting metastatic proclivity rather than a stimulator of cell growth and proliferation.

We then retrospectively analyzed preoperative DW-MRI scans of 42 PDAC patients and compared the ADC values to Blimp-1 expression determined by immunohistochemistry in resected tumor specimens. Blimp-1 expression was significantly negatively correlated with the ADC and high levels of Blimp-1 were linked to short progression-free survival (PFS). High levels of Blimp-1 were associated with a reduction of water diffusivity in the tumor (as expressed by low ADC values) which indicates high compactness of the tissue [27]. To solidify the data of Blimp-1 as a surrogate marker for intratumoral hypoxia, we evaluated the expression of VEGF, a well-established marker, which is upregulated in hypoxia, in PDAC tissue samples. We observed a significant positive correlation between Blimp-1 and VEGF expression. Similar to Blimp-1, VEGF expression showed a significant negative correlation with the ADC. Of note, we detected no significant difference in median ADC values between patients with high and low tumor-stroma ratio which is in line with a recent study by Mayer et al. [28] who found the ADC to be less suited for assessment of stroma content in PDAC than the non-Gaussian diffusion coefficient from diffusion kurtosis imaging. Moreover, Blimp-1 expression and VEGF expression were not significantly different between patients with high and low tumor-stroma ratio. A possible explanation could be that, in stroma-rich tumors (i.e., low tumor-stroma ratio), the more-impaired oxygen delivery due to abundant connective tissue fibers and the lower oxygen consumption due to lower cell density cancel one another out.

Since Blimp-1 and VEGF are closely linked to tumor hypoxia, the ADC could represent a valuable surrogate parameter for the oxygenation status of PDAC. This is in congruence with previous studies by Hompland et al. who found negative correlations of ADC values and the hypoxic fraction in melanoma xenografts [31] and human prostate cancer [32,50]. A significant association of low ADC values and hypoxia, as detected by FMISO-PET, was also reported for head and neck cancers in a recent study by Wiedenmann et al. [51] although a previous study described no significant correlation between the ADC and HIF-1α in oropharyngeal carcinoma [52].

FMISO-PET is sometimes seen as the gold standard to image tumor hypoxia and in theory directly depicts tumor oxygenation, whereas the ADC value from DW-MRI should be regarded as an indirect measure for hypoxia which quantitates tissue properties responsible for hypoxia such as cell density and stroma content [24,51]. However, DW-MRI has several potential advantages over FMISO-PET, including its wide availability, ease of use, no requirement for an intravenous contrast agent or tracer, and the absence of ionizing radiation [27]. On the other hand, limited availability, as well as the required logistics and tracer availability, can make hypoxia PET imaging demanding [24,51]. FMISO, the most widely used radiotracer for imaging hypoxia, exhibits comparatively slow pharmacokinetics and should be evaluated 2 to 4 hours after its administration [53], while the acquisition time of multi b-value DWI of the upper abdomen is usually a few minutes only [54]. Remarkably, FMISO demonstrates only minimal activity in PDAC which makes correlation with other imaging modalities for localization of the tumor necessary [55]. Other MRI techniques that are potentially suitable for the assessment of tumor hypoxia include blood oxygenation-level dependent (BOLD) MRI [56] and pattern recognition analysis of dynamic contrast-enhanced (DCE) MRI [57], but are not yet widely established in the upper abdomen.

Hypoxia is long known as one of the most important causes of radiotherapy failure [58] and has also emerged as a mediator of chemotherapy resistance in human cancers [3]. Ionizing radiation kills tumor cells through the direct and indirect generation of double-stranded breaks in the deoxyribonucleic acid (DNA). According to the oxygen fixation hypothesis, the presence of oxygen makes these damages very difficult for the cell to repair (i.e., oxygen “fixes” the damage). Therefore, DNA damage induced in the presence of hypoxia is 2.5–3 times less likely to end in cell death [59]. Hypoxia-induced chemoresistance may be mediated by several factors, including the induction of various genes controlling cell metabolism and survival [3] as well as induction of stemness [60]. Therefore, information about tumor hypoxia may help to individualize therapy concepts, e.g., by dose escalation of radiation therapy in hypoxic tumors [51] as recently reported for non-small cell lung cancer [61]. Longitudinally monitoring tumor hypoxia during the course of treatment could be beneficial as tumor hypoxia may decrease during radiochemotherapy [51,62] which could make a readjustment of treatment necessary. In this aspect, the above-mentioned advantages of DW-MRI over FMISO-PET become even more significant [51]. Future studies should be conducted to compare DW-MRI with FMISO-PET for assessment of intratumoral hypoxia in PDAC.

In conclusion, we show that DW-MRI could be used as a patient-based non-invasive surrogate for tumor hypoxia in PDAC with potential impact on personalized treatment planning.

## 4. Materials and Methods

### 4.1. Patients

The study protocol received Institutional Review Board (IRB) approval (S-044/2012, Ruprecht Karls University Heidelberg, Germany). We retrospectively searched the patient database of the Department of Diagnostic and Interventional Radiology of the University Hospital of Heidelberg for patients who had surgical resection of PDAC without neoadjuvant therapy between since March 2013 and who had a DW-MRI (with b50 = 50 s/mm^2^ and b800 = s/mm^2^) of the pancreas on the day before surgical tumor resection. Forty-two patients (19 women, 23 men, mean age 66.3 years (range 42 years to 85 years)) who matched these criteria were included in the study. Tumor-node-metastasis (TNM) staging was as follows: 19 patients had T2 tumors and 23 patients had T3 tumors. Lymph node metastases were detected in 32 patients, of whom 8 patients were classified as N1 and 26 patients as N2; 8 patients had no evidence for lymph node metastases (N0). In 2 patients, solitary distant metastasis was detected intraoperatively and resected (M1, liver metastasis in 1 patient, lymph node metastasis next to renal vessels in 1 patient). Twenty-five tumors were classified as G2 (moderately differentiated) and 17 tumors as G3 (poorly differentiated). Thirty-three tumors were located in the pancreatic head, 9 tumors were located in the pancreatic body and/or tail. Total pancreatectomy was conducted in 7 patients, pylorus-preserving Whipple procedure in 18 patients, classical Whipple procedure in 9 patients, and pancreatic left resection in 8 patients. Thirty-five out of 42 patients had adjuvant chemotherapy, either gemcitabine-based or FOLFIRINOX (folinic acid, fluorouracil, irinotecan, oxaliplatin). Progression-free survival was determined based on follow-up CT scans. Mean follow-up was 285 days (range 0 to 1988 days). Locoregional or metastatic progression was detected in 13 patients (Appendix A).

### 4.2. Diffusion-Weighted Magnetic Resonance Imaging

All DW-MRI scans were conducted on a clinical 1.5-T scanner (MAGNETOM Aera, Siemens Healthcare, Erlangen, Germany). The maximum gradient strength of the scanner is 45 mT/m. A body-phased array coil with 6 elements and a spine array coil with 24 channels were used. DW imaging was performed with a single-shot echo-planar imaging (ss-EPI) sequence in expiratory breath hold. DWI parameters were as follows: transversal orientation; 14 slices; slice thickness, 5.0 mm; distance factor, 5%; field of view (FOV), 350 mm × 248 mm; matrix size, 130 × 92; receiver bandwidth, 2262 Hz per pixel; parallel imaging technique, based on k-space (GRAPPA, generalized autocalibrating partially parallel acquisition; acceleration factor: 2); three orthogonal gradient directions. Spectral fat saturation was applied.

The Medical Imaging Interaction Toolkit Diffusion application (MITK diffusion, Version 2017.07, developed by the Division Medical Imaging Computing of the German Cancer Research Center, Heidelberg, Germany; https://www.MITK.org) was used for DWI-analysis. All PDAC lesions were independently segmented on diffusion-weighted images by two board-certified radiologists (blind) with 8 and 15 years of experience in pancreatic imaging. Segmentations encompassing the tumors were drawn on DW images. For placement of the volumes of interest (VOIs) on DWI, the anatomical outlines of the PDAC lesions were determined with the help of anatomical MR images and CT images. The signal magnitudes with diffusion weightings S(b50) and S(b800) were extracted and the ADC was computed, as described earlier by Penner et al. [63]:(1)$${\mathrm{ADC}}_{\left(50,800\right)}=\;\frac{\ln\left(S\left(b50\right)\right)-\ln\left(S\left(b800\right)\right)}{b800-b50}.$$

### 4.3. Immunohistology

Human tissue samples were provided by the tissue bank of the National Center for Tumor Diseases Heidelberg (NCT, Heidelberg, Germany) in agreement with the regulations of the tissue bank and local IRB approval (no. 206/2005). All patients gave written informed consent. Analyses were performed on whole tumor sections to overcome tumor heterogeneity. For immunohistochemical analysis, heat-induced antigen retrieval was conducted using buffer (pH 6.0 for Blimp-1, pH 9.0 for VEGF; Dako EnVision, Glostrup, Denmark). Rabbit anti-human Blimp-1 antibody was used (1:50; Cell Signaling Technology, Leiden, Netherlands). Antibody-binding of Blimp-1 was visualized with permanent magenta chromogen (Dako), of VEGF with DAB+ chromogen (Dako) and analyzed using the semi-quantitative Allred score [64]. Expected staining of Blimp-1 is nuclear, of VEGF cytoplasmatic. The avian xenograft tumors were resected, fixed in formalin, embedded in paraffin, and stained similarly to human tissue, as described above. Images were taken using the Gryphax Subra camera (Jenoptik, Jena, Germany). For confirmation of results, quantification was done and documented by ImageJ (US National Institutes of Health, Bethesda, MD, USA; https://imagej.nih.gov/). The percentages of tumor tissue and stroma were determined semiquantitatively on hematoxylin and eosin (HE) tissue [29].

### 4.4. In Vitro Experiments

#### 4.4.1. Cell Culture

The following human PDAC cell lines were used: AsPc1 (obtained from ATCC, Manassas, VA, USA) and T3M4 (European Pancreas Center Heidelberg, Heidelberg, Germany). Cells were cultivated in RPMI 1640, in presence of 10% fetal bovine serum (FBS) and 1% penicillin and streptomycin (Life Technologies GmbH, Darmstadt, Germany). Cells were incubated at a temperature of 37 °C, CO_2_ concentration of 5%, and relative humidity of 95%. For all experiments, cells were harvested in their linear growth phase.

#### 4.4.2. Treatment of Tumor Cells with Cobalt(II)Chloride

Cells were seeded in a concentration of 3 ×10^5^ / mL (T3M4) or 4 ×10^5^ / mL (AsPc1), respectively, which resulted in a density of 80 to 90% when grown overnight. The medium was then replaced, by medium either supplemented with Cobalt(II)chloride (Hypoxia group) (200 µM; Sigma-Aldrich GmbH, Taufkirchen, Germany) or sterile as water vehicle control (Lonza Group, Basel, Switzerland). After the indicated times cells were harvested for further analysis. A fraction of cells was treated with the ERK-inhibitor sc-203945 (30 µM; Santa Cruz Biotechnologies, Heidelberg, Germany). The susceptibility of PDAC for Cobalt(II)chloride induced hypoxia by upregulation of HIF-1 was documented for AsPc1, and to a lower level, also for T3M4 [47,48,65].

#### 4.4.3. Cell Lysis and Western Blot Analysis

After harvesting and lysis of cells using the radioimmunoprecipitation assay (RIPA) Lysis Buffer System (sc-24948A; Santa Cruz Biotechnologies), the total protein content of the solubilized material was assessed using the Pierce™ bicinchoninic acid (BCA) Protein Assay Kit (Thermo Fisher, Dreieich, Germany). Sodium dodecyl sulfate polyacrylamide gel electrophoresis (SDS-PAGE, 10%) was performed with a protein content of 20 µg and Prestained Rec Protein Ladder as size marker (Fisher Scientific GmbH, Schwerte, Germany).

Proteins were transferred onto a nitrocellulose membrane (0.45 μm, Bio-Rad Laboratories GmbH, Feldkirchen, Germany) for Western blotting. The membrane was blocked with 3% bovine serum albumin (BSA, Sigma-Aldrich GmbH, Taufkirchen, Germany). The following primary antibodies were applied: mouse anti-human Vinculin (V9131, Sigma-Aldrich GmbH, Taufkirchen, Germany), mouse anti-human β-tubulin (sc-58886; Santa Cruz Biotechnologies, Dallas, Texas), rabbit anti-human Blimp-1 (9115S; Cell Signaling Technology), rabbit anti-human ERK (9102S; Cell Signaling Technology), and rabbit anti-human pERK (9101S; Cell Signaling Technology). After overnight incubation at 4 °C, the following secondary antibodies were used for visualization of antibody binding: goat anti-mouse immunoglobulin G (IgG) (H+L)- horseradish peroxidase (HRP) conjugate or a goat anti-rabbit IgG (H+L)-HRP (Bio-Rad Laboratories GmbH, Feldkirchen, Germany) in 5% skim milk (T145.2; Carl Roth GmbH, Karlsruhe, Germany). Protein bands were detected by a chemiluminescence reaction (CL) and Fusion Solo S chemiluminescence imaging system (Vilber Lourmant Deutschland GmbH, Eberhardzell, Germany) and documented and quantified by ImageJ.

For each experimental condition (cell culture), respectively, 6 independent experiments were performed.

#### 4.4.4. Avian Xenograft

The general setup of the assay was previously described in detail [18]. The following modifications were applied: A digital motor breeder Type Easy 250 (J. Hemel Brutgeräte GmbH & Co. KG, Verl-Kaunitz, Germany) was used for incubation of fertilized white Leghorn chicken eggs (LSL Dieburg, Dieburg, Germany) at a humidity of 45–55% at 37.8 °C. On day 5 of embryonic development, a syringe was used to remove 3–5 mL of albumen to obtain a flat surface. A small incision was performed into the eggshell and the incision was sealed with tape. At day 8 of embryonic development, small intra-oral latex elastics (Dentaurum GmbH & Co. KG, Ispringen, Germany) were positioned on the CAM and 2 × 10^6^ tumor cells (AsPc1) in 50 µL Corning^®^ Matrigel^®^ Matrix (Corning GmbH, Wiesbaden, Germany) were deposited into the elastics. At days 13 and 15 of embryonic development, 30 µL of Cobalt(II)chloride (200µM; Sigma) were added on a filter paper, placed on the tumor to allow diffusion. Limulus Amebocyte Lysate (LAL) Reagent Water (Lonza Group, Basel, Switzerland) was employed as vehicle control. On day 16, the embryos were euthanized using Ketamin (Hameln pharma plus GmbH, Hameln, Germany) and tumor size was quantitated.

After tumor resection, tumor volumes were calculated using the following formula:(2)$$V=\frac{4}{3}*\pi *r^{3}.$$

The tumor samples were fixed in formalin and embedded in paraffin for further immunohistochemical evaluation.

### 4.5. Human in Vivo Tissue Culture

Fresh human tissue was provided by the biobank of the University Medical Center Mainz, Germany, with approval of the local IRB (No.: 2019-14390). After surgical removal, the pancreas specimens were immediately transported to the Institute of Pathology. A viable piece of tumor tissue was sampled using a biopsy punching tool (Stiefel, Biopsy Punch, GlaxoSmithKline, Brentford, UK, Ø = 6 mm) and temporarily stored in 4 °C chilled Krebs-Henseleit Buffer (Sigma-Aldrich/Merck, Darmstadt, Germany). Tissue punches were then fixed and immobilized using cyanoacrylate tissue adhesive and an agar ring. Fixed punches were cut into thin homogenous slices of 300 μm thickness using a vibrating-blade microtome VT1200 (Leica Biosystems, Nussloch, Germany) equipped with a razor blade (Wilkinson Sword, Solingen, Germany). Slices were collected in 4 °C chilled Krebs-Henseleit Buffer and randomized before distribution to the control and therapy group. The first slice and the last slice of each punch were immediately fixed in 4% buffered formalin (control slice, time point 0 h) and paraffin-embedded for histomorphological evaluation. Tissue slices were cultured on cell culture inserts (Polyethylene Terephthalate membrane, 0.4 μm pore size, Falcon, Corning, NY, USA) placed in six-well plates (Greiner Bio-One, Frickenhausen, Germany) to allow air exposition to the upper surface of the tissue slice, while the lower surface is fed by the medium through the pores of the growth membrane. Dulbecco’s modified Eagle’s medium (DMEM) (ATCC, Manassas, FL, USA) cell culture medium enriched with 10% fetal bovine serum (FBS) (HyClone, Thermo Scientific, Dreieich, Germany) as well as 1% penicillin/streptomycin (Sigma-Aldrich/Merck, Darmstadt, Germany) was used for cultivation. To achieve a better supply of oxygen, plates were put on an orbital shaker (Thermo Scientific, MaxQ2000 CO_2_ Plus, 55 rpm). Tissue slices were cultured at 37 °C in a humidified incubator under atmospheric oxygen and CO_2_ levels. The medium was changed after 1 h for an additional 24 h. Then, tissue slices were treated with a concentration of cobalt chloride (200 μM) simulating hypoxic conditions. Concentration was chosen based on cell-culture experiments. After 24 h, tissue slices were harvested and fixed in 4% buffered formalin. Tissue samples, which could not be processed immediately after harvesting, were directly frozen after slicing and stored in kryo medium (DMEM cell culture medium enriched with 20% FBS and 10% dimethyl sulfoxide (DMSO, Sigma-Aldrich/ Merck, Darmstadt, Germany)) in liquid nitrogen. For cultivation, slices were defrosted, placed on the cell culture inserts, and treated as described. In order to ensure the tumor-heterogeneity, cultivation was performed in triplicates. The formalin-fixed tissue slices were put into paraffin and cut into 2 μm sections by a microtome for morphological and immunohistochemical assessment.

### 4.6. Statistics

Statistical analyses were conducted with MedCalc (MedCalc Software, Version 19.5.6, Ostend, Belgium). Inter-rater reliability for ADC values was quantitated using the intra-class correlation coefficient (ICC) (model: two-way random). Spearman rank correlations were calculated between ADC values and Blimp-1 Allred scores. For calculation of differences between groups, normality of the data was tested with the Shapiro–Wilk test. If differences followed a normal distribution, Student’s t-test was used. In other cases, Mann–Whitney test or Wilcoxon test were used. Progression-free survival was estimated from the date of surgery to the date of disease progression (locoregional or metastatic, based on follow-up CT scans) or last follow-up. PFS curves were estimated using the Kaplan–Meier method and compared by log-rank test. All analyses were two-tailed and values of *p* < 0.05 were considered statistically significant.

## 5. Conclusions

In summary, our data show that DW-MRI qualifies as a patient-based non-invasive surrogate for tumor hypoxia in PDAC. DW-MRI could be used for identification of lesions with high hypoxic fraction which are at high risk for failure of radiochemotherapy. Moreover, due to the absence of radiation exposure, wide availability, and absence of requirement of intravenous contrast agents or radiotracers, DW-MRI is well-suited for longitudinal monitoring of tumor hypoxia during the course of treatment.

## Figures and Tables

**Figure 1 cancers-13-00089-f001:**
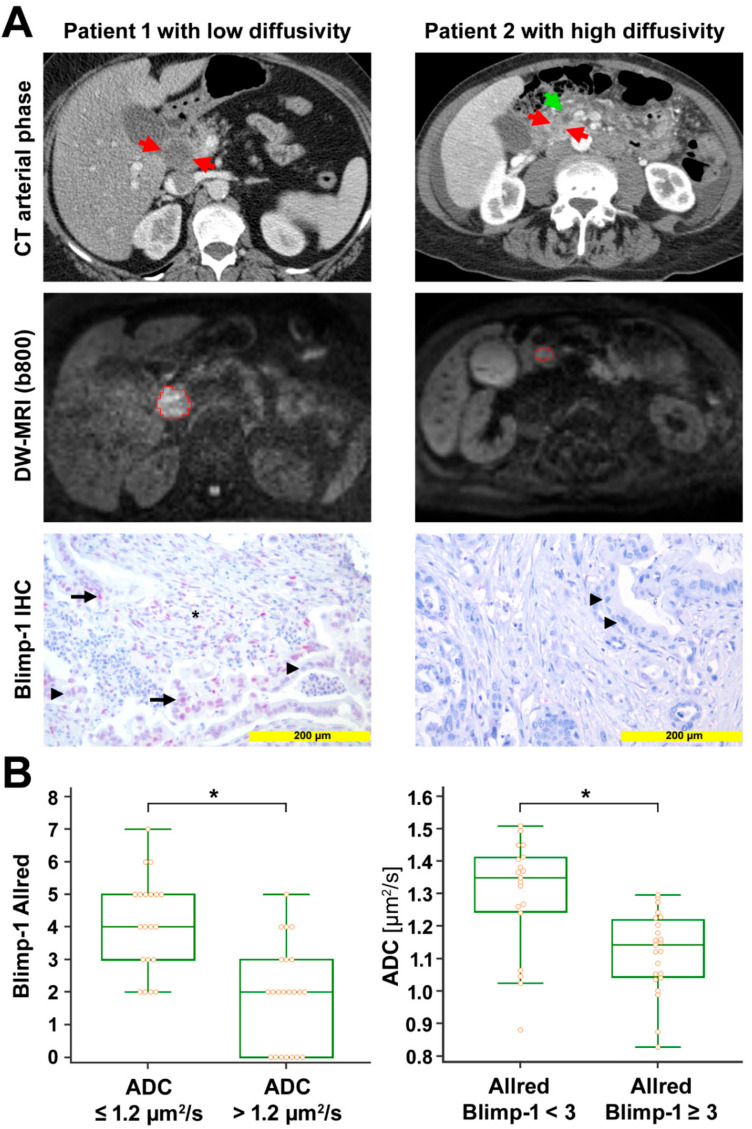
Radio-pathological correlation. (**A**) Left column: Representative images from a patient with low diffusivity. Late arterial phase computed tomography (CT) image shows the hypodense tumor in the head of the pancreas (red arrows). Diffusion-weighted image (DWI) with the volume of interest (VOI, red) from radiologist 1 encompassing the markedly hyperintense tumor (b = 800 s/mm^2^). Mean apparent diffusion coefficient (ADC) value for both radiologists was 0.875 µm^2^/s. Anti-Blimp-1 immunohistochemical staining shows pancreatic cancer tissue with partial Blimp-1 expression in moderate intensity (red nuclei; arrow); Blimp-1 Allred score was 5 in this example. Negative nuclei are marked with arrowheads. Blimp-1 infiltrated immune cells in the tumor stroma, marked by an asterisk. **Right column:** Representative images from a patient with high diffusivity. Late arterial phase CT image shows the hypodense tumor in the pancreatic head (red arrows). The pancreatic duct is dilated in the upstream parenchyma (green arrow). DWI with the VOI (red) from radiologist 1 encompassing the mildly hyperintense tumor (b = 800 s/mm^2^). Mean ADC value for both radiologists was 1.509 µm^2^/s. Anti-Blimp-1 immunohistochemical staining shows pancreatic cancer tissue without nuclear Blimp-1 (blue nuclei; arrowheads). (**B**) The box-and-whisker plots show the distribution of Blimp-1 Allred scores relative to ADC values (**left**) and the distribution of ADC values relative to Blimp-1 Allred scores (right). *, *p* < 0.05.

**Figure 2 cancers-13-00089-f002:**
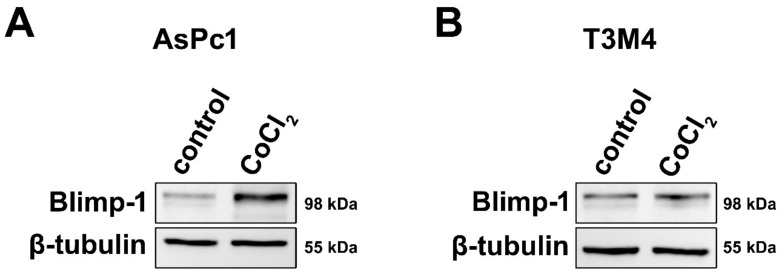
Cell culture. (**A**) Western blots showing markedly increased Blimp-1 expression in AsPc1 cells following treatment with CoCl_2_ compared to vehicle control. (**B**) Treatment with CoCl_2_ only induced a slight increase in Blimp-1 expression in T3M4 cells.

**Figure 3 cancers-13-00089-f003:**
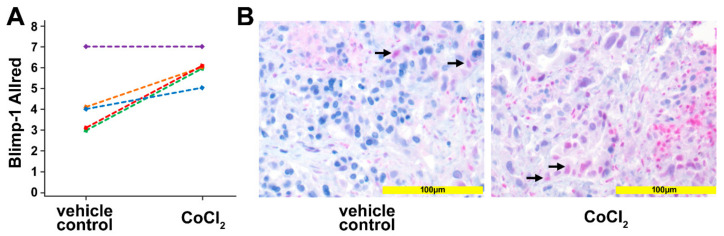
Ex vivo tissue culture. (**A**) Scoring the Blimp-1 immunohistochemical staining of human pancreatic cancer samples treated with 200 µM CoCl_2_ for 24 h. An upregulation of Blimp-1 in the nucleus was detected compared with vehicle-treated controls. Each line represents one slice. (**B**) Left: Vehicle control pancreatic cancer tissue shows scattered and weak nuclear Blimp-1 expression (arrows). Right: Pancreatic cancer tissue treated with CoCl_2_ for 24 h shows Blimp-1 expression with wider distribution and moderate staining intensity.

**Figure 4 cancers-13-00089-f004:**
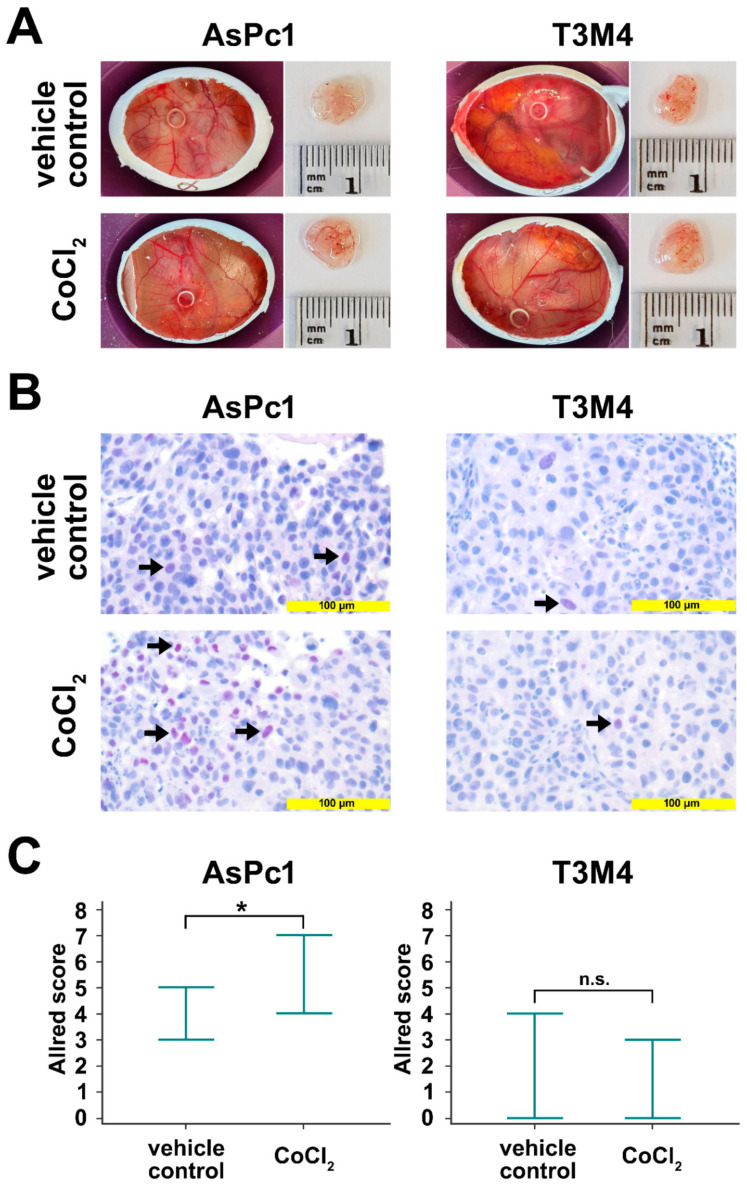
Avian xenografts model. (**A**) The three-dimensional growth of AsPc1 cells (treated, *n* = 7; untreated, *n* = 5) and T3M4 cells (treated, *n* = 4; untreated, *n* = 3) inoculated on the chorioallantoic membrane (CAM) of fertilized chicken eggs and the harvested tumors from vehicle control and after treatment with CoCl_2_ were photographed at day 17 of embryonic development. Tumor volumes and weights were not significantly different between vehicle control and CoCl_2_-treated xenograft tumors. (**B**) Anti-Blimp-1 immunohistochemical staining of vehicle control CoCl_2_-treated resected xenograft tumors. There was a significant induction of Blimp-1 expression following treatment with CoCl_2_ in AsPc1 cells, but not in T3M4 cells. (**C**) Quantification of Blimp-1 expression in xenograft tumors. Markers indicate the range of Blimp-1 Allred scores. For AsPc1 cells, Blimp-1 Allred scores were significantly higher CoCl_2_-treated than for vehicle control xenograft tumors. No significant difference in Blimp-1 Allred scores was observed for CoCl_2_-treated T3M4 xenograft tumors compared to vehicle control. n.s., not significant; *, *p* < 0.05.

## Data Availability

The numerical data sets analyzed in the current study are available from the corresponding author on reasonable request. The Digital Imaging and Communications in Medicine (DICOM) files cannot be made freely available due to privacy restrictions.

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
