# Peer review of "Restricted Water Diffusion in Diffusion-Weighted Magnetic Resonance Imaging in Pancreatic Cancer is Associated with Tumor Hypoxia"

_cancers, 2020, doi:10.3390/cancers13010089_

Round 1

Reviewer 1 Report

Mayor et al tried to establish DW-MRI as a tool to measure PDAC hypoxia by correlating water diffusion with expression of Blimp1, a hypoxia inducible gene. While this study is very interesting but the reviewer is confused why Blimp1 was chosen as a hypoxia surrogate marker since better markers such as CA9 have already been well documented.

The immunohistochemical pattern of Blimp1 looks distinct from the pattern of well-known hypoxia marker CA9, raising the question that Blimp1 might not be a good hypoxia surrogate marker. 

Indeed, based on the two PDAC models, CoCl2 only induced Blimp1 expression in one cell line, indicating that Blimp1 might not be a good hypoxia marker to PDAC. Other hypoxia markers should be tested to support the conclusion.

The regulation of Blimp1 by ERK was not correlated with the study and had no impact on the conclusion. ERK basically regulates synthesis of proteins including HIF1a. ERK inhibitors could suppress a number of proteins. I would suggest to remove ERK part, which lacks scientific stringency. 

Reviewer 2 Report

The authors conducted a retrospective study trying to examine the DW-MRI to identify the risk of treatment failure for pancreatic cancers. The findings from this study imply that DW-MRI can be used to identify pancreatic cancer lesions with high levels of hypoxia which are at high risk for treatment failure.

  1. This study is well designed and the manuscript is quite well written. The methods are adequate and statistical analysis is proper. The results justify the conclusions drawn. It would be useful to discuss the clinical significance and future work of this study.
  2. It would be better to analyze the treatment failure pattern and local control of these patients to correlate with the pre-treatment DW-MRI data.
  3. Modifications of water diffusion induced by different factors acting on the extracellular and intracellular spaces, increased cell density, edema, and fibrosis could alter signal intensity on DW images; tumor hypoxia may not the only impact factor. Does the pathological findings (e.g. fibrosis, edema) of the tumor associate with ADC values?
  4. The article showed tumor hypoxia in PDAC is reflected by DW-MRI and the tumor hypoxia correlated with treatment failure, could you provide the oncological outcome (e.g. disease-free survival or overall survival) of the 42 PDAC patients and the correlation with ADC values.
  5. As mentioned in the article, FMISO-PET is the gold standard of image study for organ or tissue hypoxia, was the data of FMISO-PET available in the study populations? Does the finding of FMISO-PET correlate with DW-MRI?
  6. In addition to BLIMP-1, could you show more hypoxia makers presented in patient samples?

     7. In fig.2, hypoxic marker HIF-1α should be analyzed to ensure the

         hypoxia condition induced by CoCl2.

Round 2

Reviewer 1 Report

The authors havre addressed the concerns.

Supplemental Figure 1 should moved to main manuscript Figure 1. 

Reviewer 2 Report

The authors have responded to my questions.